# A new approach to quantify angles and time of changes-of-direction during soccer matches

**Tomohiro Kai**[1,2], **Shin Hirai**[3], **Yuhei Anbe**[1], **Yohei Takai**[1]*

**1** National Institute of Fitness and Sports in Kanoya, Kanoya, Japan, **2** Kagoshima United FC, Kagoshima, Japan, **3** Mizuno Corporation, Suminoe-ku, Japan

* y-takai@nifs-k.ac.jp

**Data Availability Statement:** All relevant data are within the manuscript and its Supporting Information files.

**Funding:** This study was supported by a NIFS project for TASS. The funders had no role in study

## Abstract

### Background and aims

Soccer players frequently perform change-of-directions (CODs) at various speeds during matches. However, tracking systems have shown limitations to measure these efforts. Therefore, the aim of the present study was to propose a new approach to measure CODs using a local positioning system (LPS), and clarify position-related difference in profile of CODs by using the approach.

### Methods

The x- and y-coordinate data for each soccer player were measured with a local positioning system. Speed, acceleration, jerk, and direction of speed were derived from the coordinate data. Based on accelerations of above 2 m/s², the onsets and ends of CODs derived from jerk were identified (COD duration). Changes of direction of speed ($\theta_{COD}$) were determined for the corresponding period. Six collegiate male soccer players performed CODs according to 13 set angles (0–180˚; every 15˚) so that differences between $\theta_{COD}$ and set angle could be determined (Exp. 1). Relative frequency distributions of $\theta_{COD}$ and number of CODs were determined in 79 collegiate and amateur male soccer players during 9 soccer matches (Exp. 2).

### Results

In Exp. 1, $\theta_{COD}$ was positively related to set angle ($r = 0.99$). Each $\theta_{COD}$ was smaller than the corresponding set angle, and the difference became greater with increasing COD angle. In Exp. 2, The number of CODs in a match was 183 ± 39 across all positions. There were no significant position-related differences in the number of CODs. The duration of a COD was 0.89 ± 0.49 s across all positions. The relative frequency distribution of $\theta_{COD}$ revealed that the number of CODs at 0–15˚ and 105–135˚ tended to be higher than those at other angles during soccer matches. Further, $\theta_{COD}$ was affected by the speed at the onset of COD during soccer matches (Exp. 2).

design, data collection and analysis, decision to publish, or preparation of the manuscript.

**Competing interests:** The authors have declared that no competing interests exist.

## Conclusions

The current findings demonstrate that $\theta_{COD}$ derived from direction of speed and jerk may be a new indicator for evaluating COD during soccer matches.

## Introduction

Recently, coaches and sports scientists of ball sports team use tracking technologies for designing training program and players' condition, because it has become easy to measure players' locations in ball sports using global positioning system (GPS) and local positioning system (LPS) [1, 2]. Many earlier studies have demonstrated the validity of position, speed and acceleration data obtained from such a tracking technology, compared to a 3D motion capture system, radar/laser guns and timing gates [3–9]. Further, LPS is superior to GPS, whereas each system has a certain validity [10]. These systems are capable of measuring players' coordinates, and for quantifying players' acceleration, distance and numbers of actions in relation to speed derived from time-motion analysis. These parameters are predictors of match outcome and periodization on daily training [11, 12].

Besides time-motion analysis, the parameters obtained from tracking technologies are applied to quantify CODs locomotion in sport-specific course and small-sided games [3–6, 9, 10]. In the earlier studies, various type of courses which angles of CODs are predetermined are set [3, 5, 6, 9, 10], and small court is used [4]. Although the earlier findings demonstrate the magnitude of speed and acceleration, the experimental design is not real soccer matches. During soccer matches, players perform changes-of-direction (CODs) during locomotion [13]. Notational analysis has revealed that 30% of all actions during English FA Premier League play were CODs (e.g., forward, lateral and backward running) [14]. However, quantifying relevant data, such as the number and type of actions, is a lengthy process [15], and notational analysis may be arbitrary [16]. Therefore, a convenient analytic method to quantify CODs during soccer matches is needed. Fitzpatrick et al. [2] demonstrated that, in the English U-18 Premier League, the direction of players' locomotion at a speed of 6.67 m/s or more ranged from 0˚ to 30˚ by using GPS. This suggests that during matches, youth soccer players move close to a straight line at relatively high speeds. Further, an earlier study of soccer matches revealed that greater distances are covered at moderate speeds of 3.89 to 5.28 m/s than at high-intensity speeds of 5.28 to 6.39 m/s [17]. Although soccer players frequently perform CODs at various speeds during matches, to the best of our knowledge, little information is available concerning COD profiles during soccer matches in relation to speed by using LPS.

Force is theoretically the product of mass and acceleration. Acceleration can be useful in describing a player's physical load during soccer matches. Dalen et al. [1] demonstrated that position-related differences in the number of accelerations ($>2$ m/s$^2$) was found in the first division of the Norwegian league. When a player changes direction of locomotion, he exerts a certain force against the ground. At the same time, a certain level of acceleration is produced, and then the direction of the player's locomotion changes, and the speed and/or the direction of speed changes [3]. Jerk, which is derived by differentiating acceleration by time, is used to detect the onset of human joint movement and the magnitude of the movement [18]. Therefore, jerk should be useful in identifying the onset and end of a COD for a given acceleration, and the change in direction of speed should correspond to the direction of the COD.

During professional soccer matches, position-related differences in acceleration are found, indicating that side midfielders and defenders accelerate more often than other positions [1].

Further, midfielders, relative to other positions, performed fewer CODs of 90 degrees or less [14]. Considering these findings, profile of CODs would differ among positions. This study thus proposes a new approach that uses direction of speed and jerk in quantifying CODs of soccer players during soccer matches, and to clarify position-related difference in profile of CODs by using the approach.

## Materials and methods

### Experimental design

This investigation was conducted according to the Declaration of Helsinki and approved by the local Ethics Committee for human experimentation (#11–101). All participants were informed of the experimental procedures and possible risks of the measurements beforehand. Oral informed consent was obtained from each subject before each match.

This study consisted of two experiments for quantifying CODs during soccer matches in relation to various speeds. In both experiments, players' locations were measured with LPS. In the first experiment 1 (Exp. 1), male soccer players performed CODs in directions determined by 13 set angles (every 15˚, ranging from 0˚ to 180˚) while jogging at speeds of approximately 1.0 and 3.0 m/s. After the player turned at the determined location, he ran through a gate set at a distance of 2 m from the corresponding location. The participants were asked to perform CODs as fast and quickly as possible when they turned in a given direction. S1 Fig presents typical trajectory data and kinematic data (to be discussed below) in each angle for one player.

In the second experiment 2 (Exp. 2), data were collected from 9 official soccer matches in Division 1 of a regional collegiate male soccer league and the Division 5 of a regional amateur soccer league for collegiate and amateur soccer players. Data were analyzed for the players who played for 90 min.

### Participants

In the Exp. 1, six collegiate male soccer players (age, 21.0 ± 1.5 years, height, 172.8 ± 6.1 cm, body mass, 66.8 ± 9.2 kg; means ± SDs) participated in Exp. 1. They were field players, and belonged in the same team competing in a national university league in Japan, and had experience of competitive soccer training for >9 years. They had participated in regular soccer-specific training for more than five days (>1.5 hours/day) per week.

Seventy-nine collegiate and amateur male soccer players (23.0 ± 4.1 years, 173.9 ± 5.1 cm, 67.5 ± 6.2 kg) involved in Exp. 2, and got in the official soccer matches in in Division 1 of a regional collegiate male soccer league or the Division 5 of a regional amateur soccer league. The number of players in each position was as follows; 23 players for central backs (CB, 23.4 ± 4.7 years, 177.4 ± 5.0 cm, 71.2 ± 5.7 kg), 16 players for side backs (SB, 22.6 ± 3.4 years, 171.7 ± 3.9 cm, 65.2 ± 4.7 kg), 15 players for central midfielders (CMF, 22.2 ± 2.9 years, 172.3 ± 5.0 cm, 64.8 ± 6.3 kg), 14 players for side midfielders (SMF, 23.7 ± 5.0 years, 172.3 ± 4.2 cm, 65.9 ± 4.9 kg), and 11 players for forwards (FW, 21.8 ± 3.7 years, 174.3 ± 4.5 cm, 69.8 ± 5.9 kg), respectively. Goalkeepers were excluded from data analysis.

All participants involved in Exp. 1 and 2 were free of cardiovascular, metabolic, and immunologic disorders and/or orthopedic abnormalities, and were not using any medications that affected their muscular function and size. All study participants provided informed consent, and the study design was approved by an ethics review board (the Ethics Committee in National Institute of Fitness and Sports in Kanoya for human experimentation (#11–101)).

## Players' coordinate data

X- and y-coordinate data for each player were measured with LPS (ZXY Sports Tracking, Chyronhego, New York, USA) at a sampling frequency of 20 Hz. A belt with a sensor (approx. 20 g) under their uniform were attached for each player. In Exp.2, three examiners helped to wear the sensor for starters before the start of soccer matches. Players were instructed to take off the belt if they felt uncomfortable during matches. The data obtained were processed as described below using Matlab (Mathworks ver. 2018b, New York, USA).

**1 Kinematic data and filtering process.** To obtain the smoothed time-series data for jerk, the time-series data of x- and y-coordinates were processed by a second-order Butterworth low-pass filter employing a zero phase lag before analysis. To identify the appropriate cutoff frequency for the low-pass filter, we repeated the filtering process at every 1 Hz from 1 Hz to 6 Hz. The time-series data for jerk with and without the filtering process are presented in S2 Fig. As shown in S2 Fig, the use of cutoff frequencies of 3 Hz to 6 Hz resulted in noise in the smoothed time-series data, while cutoff frequencies of <2 Hz produced less noise. Thus a 2 Hz cutoff frequency was adopted in this study.

**2 Kinematic data for each player.** To determine players' velocity, displacement from (*t-1*) to (*t+1*) was defined as (x(t+1)-x(t-1), y(t+1)-y(t-1)) of the smoothed coordinate data for each player. Player speed of players ($|V(t)|$) in m/s was calculated by differentiating the displacement by time. Player's acceleration ($|A(t)|$) in m/s$^2$ was derived by differentiating $|V(t)|$ by time. Finally, jerk ($j(t)$) in m/s$^3$ was calculated by differentiating $|A(t)|$ by time.

$$|V(t)| = \sqrt{(x(t+1) - x(t-1))^2 + (y(t+1) - y(t-1))^2}/\Delta t$$

$$|A(t)| = \sqrt{(|V(t+1)| - |V(t-1)|)^2}/\Delta t$$

$$j(t) = \sqrt{(|A(t+1)| - |A(t-1)|)^2}/\Delta t$$

where $\Delta t$ is the time interval of sampling (0.05 s) and $t = 2, 3, \ldots, $n-1. An example of typical kinematic data for one player is presented in S1 Fig.

**3 Direction of speed.** Angle of a COD was derived from the direction of speed as follows:

$$\text{Direction of speed (deg)} = \tan^{-1}\left|\frac{y(t+1) - y(t-1)}{x(t+1) - x(t-1)}\right|, \text{ where } t = 2, 3, \ldots, \text{n}-1.$$

To determine the onset and end of the direction of speed, the time point above 2 m/s$^2$ [1] and the local maximum value were detected from the time-series data of $|A|$. For the corresponding points, the time points at which $j$ changed from negative to positive and from positive to negative were identified and were defined as the onset and end of a COD, respectively. Change in direction of speed for the corresponding period was defined as the COD angle ($\theta_{COD}$). In this study, a positive value means that a player turned towards the right side, and a negative value means that a player turned towards the left side. Fig 1 summarizes the method of analysis of $\theta_{COD}$.

## Data analysis

In Exp. 1, we estimated the differences between the set angles and $\theta_{COD}$. In Exp. 2, the relative frequency distributions of $\theta_{COD}$ within 15˚ bin ranges (0–15˚, 15–30˚, 30–45˚, 45–60˚, 60–75˚, 75–90˚, 90–105˚, 105–120˚, 120–135˚, 135–150˚, 150–165˚, 165–180˚, and >180˚) were

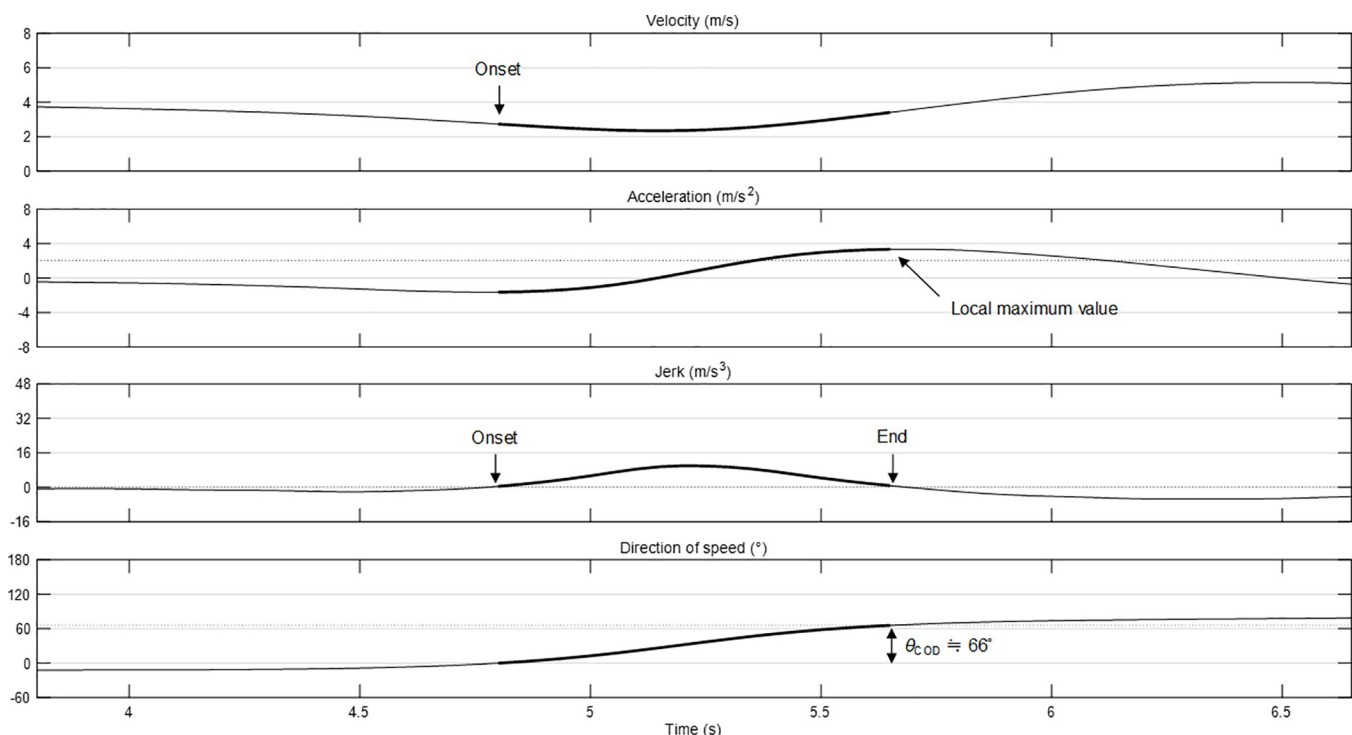

**Fig 1. Analysis method for $\theta_{COD}$.** Dotted line in line plot of acceleration indicates a threshold of acceleration (2 m/s$^2$).

calculated for each position. The number of CODs and analytical durations were also determined. Further, we examined the relationship between the speed at the onset of a COD and $\theta_{COD}$ during soccer matches.

## Statistical analysis

Descriptive data are expressed as means and standard deviations (SDs). In Exp. 1, a one-way analysis of variance (ANOVA) was used to test differences in $\theta_{COD}$ for all combinations of set angles. Pearson's product-moment correlation coefficients ($r$) were estimated for the relationships between $\theta_{COD}$ and set angles. In Exp. 2, skewness and kurtosis were used to test whether the relative frequency distributions of $\theta_{COD}$ were normally distributed, according to the method of Yokoyama [19]. A one-way ANOVA was used to test position-related differences in number of CODs and duration of the analytical period. All statistical procedures were conducted with SPSS statistical software (SPSS 25.0, IBM, New York, USA). The significance level was set at 0.05.

## Results

### Differences between $\theta_{COD}$ and the set angles (Exp. 1)

Table 1 presents descriptive data for $\theta_{COD}$ at each set angle and the differences between set angles and $\theta_{COD}$. $\theta_{COD}$ increased with increasing set angles. Error values between the set angles and $\theta_{COD}$ also increased with increasing set angles. Duration of the analytical period was $1.04 \pm 0.62$ s. $\theta_{COD}$ was positively related to set angles ($r = 0.99$), as shown in Fig 2.

**Table 1. Descriptive data on $\theta_{\text{COD}}$ and the difference between each $\theta_{\text{COD}}$ and set angle.**

| Angle of CODs | $\theta_{\text{COD}}$ [a] | Dif |
|---|---|---|
| 0˚ | -3.2 ± 3.2 | 3.2 ± 3.2 [b] |
| 15˚ | 12.2 ± 2.7 | 2.7 ± 2.7 [b] |
| 30˚ | 23.4 ± 1.6 | 6.6 ± 1.6 [c] |
| 45˚ | 35.4 ± 2.8 | 9.4 ± 2.9 [d] |
| 60˚ | 45.9 ± 4.1 | 14.1 ± 4.1 [e] |
| 75˚ | 57.2 ± 5.6 | 17.6 ± 5.4 [f] |
| 90˚ | 68.1 ± 8.7 | 21.5 ± 8.5 [g] |
| 105˚ | 77.2 ± 5.1 | 27.8 ± 5.1 [h] |
| 120˚ | 89.2 ± 5.6 | 30.8 ± 5.6 |
| 135˚ | 102.5 ± 3.9 | 31.8 ± 4.6 |
| 150˚ | 112.0 ± 4.1 | 37.8 ± 4.0 [i] |
| 165˚ | 131.4 ± 4.9 | 33.4 ± 5.2 [i] |
| 180˚ | 155.0 ± 7.5 | 24.6 ± 7.7 |

Values are means and SDs. COD, change of direction; a, significant difference with all combinations; b, significant difference between the corresponding angle and set angles above 60˚; c, significant difference between corresponding angle and set angles above 75˚; d, significant difference between corresponding angle and set angles above 90˚; e, significant difference between corresponding angle and set angles above 105˚; f, significant difference between corresponding angle and set angles above 105˚, except for 180˚; g, significant difference between corresponding angle and set angles above 120˚, except for 180˚; h, significant difference between corresponding angle and 150˚; i, significant difference between corresponding angle and 180˚.

## CODs profiles during soccer matches (Exp. 2)

Table 2 presents descriptive data on the number of CODs during the soccer matches. The number of CODs in a match was 183 ± 39 across all positions. There were no significant position-related differences in the number of CODs. The duration of a COD was 0.89 ± 0.49 s across all positions.

Fig 3 shows the relative frequency distributions of $\theta_{\text{COD}}$ during the soccer matches. The values of skewness indicate that the distributions of $\theta_{\text{COD}}$ were symmetric for all positions, except for SB. The skewness of SB was negative, indicating that the distribution was skewed to the left. The kurtosis was platykurtic for positions other than FW. The distribution in FW revealed leptokurtic. As seen in Fig 3, the relative frequency of CODs at 0–15˚ and 105–135˚ tended to be higher than that of CODs at other angles. Table 3 provides descriptive data on the number of CODs per match in each bin.

The speed at the onset of a COD was 1.36 ± 0.96 m/s, ranging from 0.01 m/s to 6.99 m/s. As seen in Fig 4, players executed CODs in different directions at relatively lower speeds of <5 m/s, but CODs in limited directions (-30˚-30˚) occurred at higher speeds of >5 m/s.

## Discussion

This study aims to propose a new approach that uses direction of speed and jerk in quantifying CODs of soccer players during soccer matches, and to clarify position-related difference in profile of CODs by using the approach. As the results, change in direction of speed, $\theta_{\text{COD}}$, which was derived from direction of speed and jerk, increased with increasing set angles of the predetermined course (Exp. 1). Further, the relative frequency of $\theta_{\text{COD}}$ during soccer matches revealed a platykurtic distribution in positions other than FW, but, in that of FW, the distribution was leptokurtic (Exp. 2). Therefore, the $\theta_{\text{COD}}$ proposed in this study may be an index of

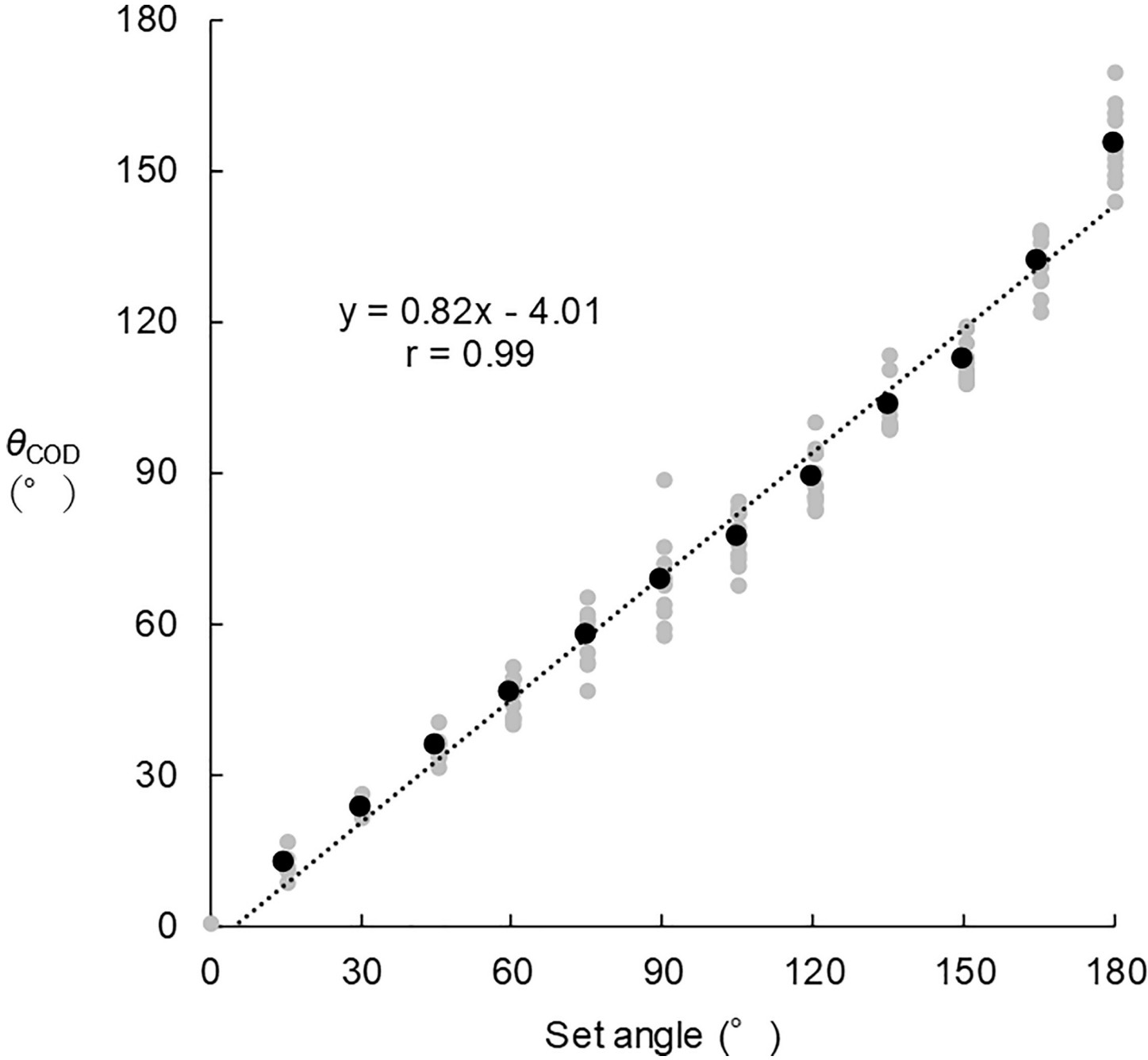

**Fig 2. Relationship between $\theta_{COD}$ and set angles.** Black dotted plot indicates the mean value of each set angle. Grey dotted plots indicate individual's data within each set angle. Dotted line indicates approximate line.

angle of CODs, and profile of CODs during soccer matches obtained from $\theta_{COD}$ may be position-specific.

In Exp.1, each $\theta_{COD}$ was smaller than the set angle, and the difference between each set angle and $\theta_{COD}$ became larger as the angle increased. This may have been due to a difference in the set course and the trajectory of center of mass (COM) of a player's body. For example, in the COD at 90 deg, a player moved in an arc, rather than at a right angle, as seen in S1 Fig.

**Table 2. Descriptive data on number of CODs, analytical duration, skewness and kurtosis of frequency distribution.**

|  | Number per match (times) | Duration per COD (s) | Skewness | Kurtosis |
| --- | --- | --- | --- | --- |
| **All** | 183 ± 39 | 0.89 ± 0.49 | -1.50 | -31.50 [b] |
| **CB** | 175 ± 38 | 0.88 ± 0.49 [a] | 0.34 | -17.91 [b] |
| **SB** | 183 ± 43 | 0.90 ± 0.50 | -2.20 [b] | -13.87 [b] |
| **CMF** | 196 ± 38 | 0.90 ± 0.48 | -0.11 | -14.59 [b] |
| **SMF** | 195 ± 36 | 0.89 ± 0.50 [a] | -1.00 | -9.54 [b] |
| **FW** | 173 ± 39 | 0.93 ± 0.50 | -0.56 | 13.22 [b] |

Values are means and SDs. All, all positions; CB, center backs; SB, side backs; CMF, central midfielders; SMF, side midfielders; FW, forwards

a, Significant difference in the measured variable between FW and other positions

b, Significant different from normal distribution

The direction of COM during change-of-direction running differed from the set angle, and the angle for COM was smaller than the set angle [20–22]. Suzuki et al. [20] revealed that the difference was 2° at 30°, 8° at 60°, and 12° at 90°, and the difference increased with COD angles. In this study, the corresponding values were 6.6°, 14.1°, and 21.5°, respectively, also becoming greater as COD angle increased.

The mean values of analytical duration were 1.04 s for Exp. 1, and 0.89 s for Exp. 2. Granero-Gil et al. [23] have defined change-of-direction locomotion as curvilinear locomotion that lasts more than 0.8 s, and they attempted to detect CODs during soccer matches by using an inertial sensor. The analytical durations in the present study were close to this definition, providing support for the threshold reported by Granero-Gil et al. [23]. On the other hand, the

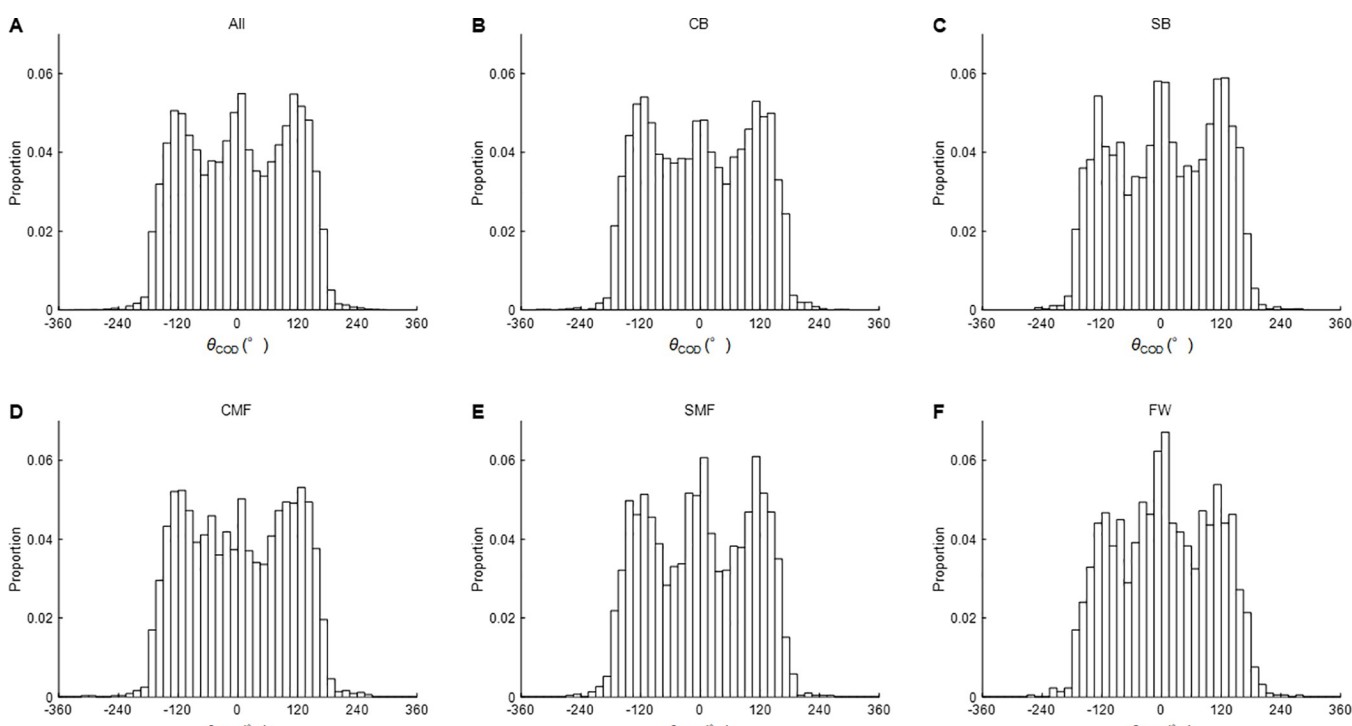

**Fig 3. Relative frequency distribution of $\theta_{COD}$ per match.** Each bin is set at every 15°. A: all positions, B: center backs (CB), C: side backs (SB), D: central midfielders (CMF), E: side midfielders (SMF), F: forwards (FW).

**Table 3. Descriptive data on number of CODs per match in each bin range.**

|  | CB | SB | CMF | SMF | FW |
|---|---|---|---|---|---|
| **0–15˚** | 16.7 ± 5.5 | 21.3 ± 7 | 17.2 ± 5.1 | 21.8 ± 7.5 | 22.4 ± 5.9 |
| **15–30˚** | 13.6 ± 4.9 | 15.5 ± 5.5 | 15.5 ± 4.5 | 18.1 ± 5.4 | 15.6 ± 4.5 |
| **30–45˚** | 13.1 ± 4.0 | 12.4 ± 4.6 | 13.7 ± 4.0 | 12.8 ± 4.9 | 15.8 ± 4.5 |
| **45–60˚** | 12.2 ± 3.8 | 12.9 ± 4.6 | 15.6 ± 4.5 | 12.7 ± 2.9 | 13.4 ± 4.5 |
| **60–75˚** | 13.5 ± 4.5 | 11.8 ± 4.7 | 16.1 ± 4.1 | 12.9 ± 3.8 | 10.6 ± 2.8 |
| **75–90˚** | 14.1 ± 4.9 | 14.8 ± 5.0 | 16.9 ± 6.2 | 14.9 ± 4.4 | 15.9 ± 6 |
| **90–105˚** | 16.3 ± 6.4 | 15.9 ± 4.9 | 18.9 ± 4.3 | 18.0 ± 4.4 | 14.2 ± 6.2 |
| **105–120˚** | 18.6 ± 5.9 | 18.4 ± 6.4 | 19.9 ± 7.1 | 21.9 ± 6.3 | 17.4 ± 6.4 |
| **120–135˚** | 17.7 ± 6.1 | 20.8 ± 6.7 | 20.6 ± 7.3 | 19.1 ± 6 | 15.2 ± 5.1 |
| **135–150˚** | 16.4 ± 5.7 | 15.6 ± 5.4 | 18.2 ± 6.6 | 18.8 ± 6.8 | 13.7 ± 5.0 |
| **150–165˚** | 11.8 ± 4.5 | 14.2 ± 5.9 | 13.2 ± 3.6 | 13.1 ± 4.6 | 8.8 ± 4.5 |
| **165–180˚** | 8.0 ± 3.3 | 7.3 ± 3.6 | 7.2 ± 3.5 | 7.2 ± 3.5 | 6.6 ± 2.8 |
| **≥180˚** | 2.7 ± 1.9 | 2.9 ± 2.1 | 3.1 ± 1.9 | 3.5 ± 2.2 | 3.3 ± 2.2 |

Values are means and SDs. All, all positions; CB, center backs; SB, side backs; CMF, central midfielders; SMF, side midfielders; FW, forwards.

number of CODs in the present matches was approx. 183, lower than the corresponding value (471 times) reported by Granero-Gil et al. [23]. This discrepancy may be due to differences in the method of analysis. In this study, $\theta_{COD}$ was estimated based on acceleration above 2 m/s$^2$ [1], while the earlier study used the definition above to identify changing-of-direction locomotion [23]. Another reason may be due to be the technical error of acceleration measured by

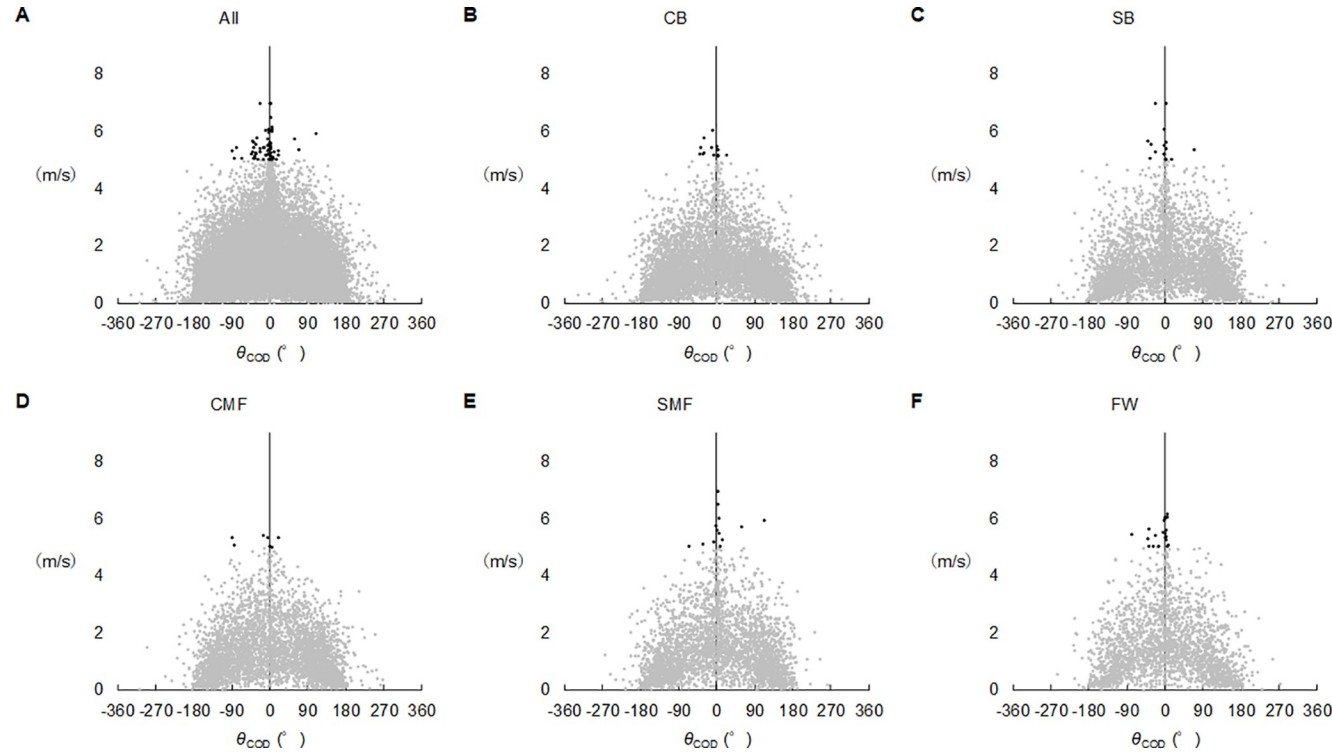

**Fig 4. Relationships between $\theta_{COD}$ and speed at onset of changing of direction (COD).** Black filled circle, the speed of more than 5 m/s; Grey filled circle, the speed below 5 m/s. A: all positions, B: center backs (CB), C: side backs (SB), D: central midfielders (CMF), E: side midfielders (SMF), F: forwards (FW).

LPS. In fact, Linke et al. [4] have demonstrated that root mean square error of acceleration in sport-specific course ranged from 0.49 m/s$^2$ to 1.34 m/s$^2$.

The degree of skewness demonstrated that $\theta_{COD}$ during soccer matches was distributed symmetrically, except for players in the side back position. This indicates that no laterality of angles of CODs may be found for amateur soccer players. On the other hand, the asymmetric distribution of $\theta_{COD}$ for SB may have been due to inter-individual differences in the distribution of $\theta_{COD}$ among SBs. In fact, the distribution of $\theta_{COD}$ was asymmetric distribution for only one SB player (skewness: -2.85) but was symmetric for the remaining SB players (skewness: -1.91 to 0.65). To this point, further investigation of a larger sample size is needed. In positions other than FW, the kurtosis analysis revealed a platykurtic distribution of $\theta_{COD}$ during soccer matches, indicating that soccer players changed direction at various angles during soccer matches. As seen in Fig 3, relative frequency of COD angles around 120 deg was more, regardless of left or right. Bloomfield et al. found that the number of CODs within 0˚-90˚ accounted for more than 80% of the total number of CODs [14]. The corresponding value in this study was approx. 50%. The discrepancy may be attributable to different analysis methods. For FW, on the other hand, the distribution of $\theta_{COD}$ during soccer matches was leptokurtic, indicating that FW may perform CODs with narrower angle than other positions.

During soccer matches, $\theta_{COD}$ ranged from 0˚ to 30˚ when speed at onset of a COD was relatively high (>5 m/s) (Fig 4). Kai et al. [24] revealed that the trajectory of players above 5 m/s was similar to liner locomotion. Fitzpatrick et al. [2] also demonstrated that direction of locomotion at speeds of above 6.7 m/s ranged from 0˚ to 30˚. Propulsive force decreases with increasing running speed [25], implying that there is less space to accelerate the player's body at a given high speed. Taken together, this evidence suggests that soccer players perform straight runs or CODs with a narrow direction angle (<30˚) at relatively high speeds.

There are some limitations in this study. Firstly, LPS are limited in high intensity effort such as high speed straight running and CODs [3, 4, 6, 9]. During sport-specific course and small-sided games, the root mean square errors of instant speed over 4.17 m/s range from 0.34 m/s to 0.39 m/s [4]. However, the values may not be enough to change relationships between instant speed at onset of CODs and angles of CODs (Fig 4). Secondly, the approach used in this study cannot be used to determine direction of a body. For example, if a player runs backward in the opposite direction immediately after he moves in a straight run, the locomotion is estimated as a COD with a 180˚ turn. Further investigation of this point is needed. Thirdly, parameters derived from LPS may be influenced by measurement condition and experimental protocol [26], although the validity of tracking systems is shown in earlier studies abovementioned [3–9]. Unfortunately, we have the relevant data, and further investigation is needed in this point.

In practical application, this study demonstrated that the relative distribution of $\theta_{COD}$ was position-specific, and $\theta_{COD}$ was affected by speed at the onset of the COD during soccer matches. To design regular soccer training that meet physical demands for each position, coaches and strength conditioners for soccer players have to know what kind of locomotion is taking place during soccer matches. Considering the current findings, the players of positions other than FW need to perform CODs toward various direction at relatively low speed (< 5 m/s). For FW, however, it's better to perform CODs toward narrow angle (< 30 deg) at relatively high speed (> 5 m/s). Thus, the current findings may be useful information to achieve the principle of training specificity for soccer.

## Conclusion

This study proposed a new approach to quantifying angle of CODs ($\theta_{COD}$) during soccer matches by using direction of speed and jerk. As the results, $\theta_{COD}$ increases with increasing

the predetermined set angle, although $\theta_{\mathrm{COD}}$ was smaller than the predetermined set angle. Further, the relative frequency of $\theta_{\mathrm{COD}}$ derived from the proposed approach revealed position-specific, and $\theta_{\mathrm{COD}}$ was affected by speed at the onset of the COD during soccer matches. The current findings suggest that the approach proposed in this study may be useful to quantify angle of CODs during soccer matches.

## Supporting information

**S1 Fig. An example of typical trajectory data and kinematic data (speed, acceleration, jerk and direction of speed) in each angle when one player performed CODs in directions determined by 13 set angles (every 15˚, ranging from 0˚ to 180˚).** A: 0˚, B: 15˚, C: 30˚, D: 45˚, E: 60˚, F: 75˚, G: 90˚, H: 105˚, I: 120˚, J: 135˚, K: 150˚, L: 165˚, M: 180˚. A bold line overlapped in line plot indicates an analytical period.
(ZIP)

**S2 Fig. The time-series data for jerk with and without the filtering process are presented.** The use of cutoff frequencies of 3–6 Hz resulted in noise in the smoothed time-series data, while cutoff frequencies <2 Hz produced less noise. Thus, a 2 Hz cutoff frequency was adopted.
(TIF)

**S1 Data.**
(XLSX)

## Author Contributions

**Conceptualization:** Yohei Takai.

**Data curation:** Tomohiro Kai, Yuhei Anbe.

**Formal analysis:** Tomohiro Kai.

**Investigation:** Yuhei Anbe.

**Methodology:** Shin Hirai, Yohei Takai.

**Project administration:** Yohei Takai.

**Writing – original draft:** Tomohiro Kai.

**Writing – review & editing:** Shin Hirai, Yohei Takai.

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
