## [Decision Letter · Decision Letter 0]

24 Feb 2021

PONE-D-21-04502

A new approach to quantify changes-of-direction locomotion during soccer matches

PLOS ONE

Dear Dr. Takai,

Thank you for submitting your manuscript to PLOS ONE. After careful consideration, we feel that it has merit but does not fully meet PLOS ONE’s publication criteria as it currently stands. Therefore, we invite you to submit a revised version of the manuscript that addresses the points raised during the review process.

After consideration of both reviewers, major revisions should be conducted to improve methods and results. I suggest to strictly follow both reviewers recommendations.

We look forward to receiving your revised manuscript.

Kind regards,

Filipe Manuel Clemente, PhD

Academic Editor

PLOS ONE

Journal Requirements:

"255 This study was supported by a NIFS project for TASS."

We note that one or more of the authors are employed by a commercial company: name of commercial company.

(2) Please also provide an updated Competing Interests Statement declaring this commercial affiliation along with any other relevant declarations relating to employment, consultancy, patents, products in development, or marketed products, etc.  

Please respond by return email with an updated Funding Statement and Competing Interests Statement and we will change the online submission form on your behalf.

4.. Please amend the manuscript submission data (via Edit Submission) to include author Yuhei Anbe.

Reviewers' comments:

Reviewer's Responses to Questions

**Comments to the Author**

1. Is the manuscript technically sound, and do the data support the conclusions?

Reviewer #1: Yes

Reviewer #2: Partly

2. Has the statistical analysis been performed appropriately and rigorously? 

Reviewer #1: Yes

Reviewer #2: Yes

3. Have the authors made all data underlying the findings in their manuscript fully available?

Reviewer #1: Yes

Reviewer #2: Yes

4. Is the manuscript presented in an intelligible fashion and written in standard English?

Reviewer #1: Yes

Reviewer #2: Yes

5. Review Comments to the Author

Reviewer #1: Dear authors and editor,

It is a pleasure the opportunity to act as a reviewer for PLOS ONE. So, thank you very much for it. I have written my comments into two sections, the general comments with the main three ideas, and specific comments. With them, I hope to help the authors to improve the manuscript, encouraging them to perform due to the relevance of this topic.

General comments

- The lack of a deep literature review makes that introduction and discussion are overviews, but not further contextualization of “state of the art” or discussion of this topic.

- I am worried with the sampling rate considered. In sport literature, two systematic reviews have recently highlighted that tracking technologies are limited in high intensity short efforts such as CODs [1,2]. However, the authors considered 20 Hz for data raw and then they down sampling. Are these sampling rates suitable for this analysis? or is it a limitation?

- In my opinion, the results section shows a deep analysis, that does not meet with the rest of the article. The results showed:

1. The difference between COD and set angle.

2. Number and duration of COD during matches.

3. Number of CODs per playing position.

Regarding to the results 1 and 2, the aim should be: the validity of a new approach to set angles and duration of CODs. These aims are contextualized and fit with the rest of the sections (introduction and discussion), however, the reason to present data per playing position is not contextualized. Why do the authors consider it?

Specific comments

Title:

- Re-consider changing “locomotion” by “set angles and time of CODs”.

Abstract:

- “CODs” is not defined the first time in it was mentioned.

- The “background” of abstract not contextualize the main objective. It should be something like: “Soccer players frequently change of directions (CODs) at various speeds during matches”. However, tracking systems have shown limitations to measure these efforts. Therefore, the aim of the present study was to propose a new approach to measure CODs using a local positioning system (LPS).”

Introduction:

- The introduction lacks from relevant literature, making non suitable the contextualization of the objective. I suggest:

o Paragraph 1: Explain the importance of change of directions. Maybe, different studies performed using PCA are suitable option to highlight the importance of change of directions and other high-intensity short efforts [3]. (The idea has been written, but in my opinion, further information is needed).

o Paragraph 2: It should explain:

What are the tracking technologies (the main basis to use the new proposal that the authors presented).

What it has been found about tracking systems and change of directions.

What has it been the main problem of accuracy in this regard (e.g sampling rate).

o Paragraph 3: it should explain “the state of the art” about all studies published in this topic (LPS and change of direction). Consider, at least, these references: [4–8]. Following this recently publicized systematic review [1], these all the validity proposals using LPS in sport setting.

*In general, the introduction has interesting ideas, but it should be deeply rewritten.

Methodology:

Separate “participants” from “experimental design”.

Participants:

Add:

- Where do the players selected? What inclusion/exclusion criteria were considered?

- Participants characteristics.

- See any scale for risk of bias, and follow it.

Experimental design:

- Line 77-78: CODs in directions determined by 13 set angles. My question is: 3 or 13 set angles?

Players´ coordinate data:

- Sampling frequency: the authors mentioned 20 Hz (and less with filtering processed). However, the use of 20 Hz has not been enough for high intensity short efforts using LPS. Why do the authors think that in this study could be suitable sampling frequency? Could be a limitation? What was the criteria to use these sampling rates?

- The authors mentioned that an LPS was used, but, several factors could affect the outcome that they could not be related with the proposed approach. Therefore, a higher precision about “the use of technology” should be made, mainly to avoid the following principle: “A good experimental design is one in which the only explanation for the change in the dependent variable is due to the treatment applied”. I suggest the use of recently published survey [9].

Results

The results are well-conducted, but they have different ways ((i) the difference between COD and set angle, (ii) number and duration of COD during matches, and (iii) number of CODs per playing position)) that are not reflected in the rest of the article. In my opinion, the article should be adapted to performed results (see general comments).

Consider explain the three protocols in the abstract.

Discussion:

- Re-write the discussion, considering the suggested articles for the introduction (paragraph 3).

- Reconsider separate a discussion for each of the results presented: (i) the difference between COD and set angle, (ii) number and duration of COD during matches, and (iii) number of CODs per playing position. If all of them are relevant for the aim of this study (see general comments).

- Add limitations.

Conclusion:

See the comments mentioned above. If following the results the aim was different folds, the conclusions should have different folds too.

References:

A further revision of the literature is needed. Different articles have been published about the use of LPS to assess CODs, and some of them were not considered.

Bibliography:

1. Rico-González M, Arcos AL, Clemente FM, Rojas-Valverde D, Pino-Ortega J. Accuracy and Reliability of Local Positioning Systems for Measuring Sport Movement Patterns in Stadium-Scale: A Systematic Review. Applied Sciences. 2020;10:5994.

2. Pino-Ortega J, Oliva-Lozano JM, Gantois P, Nakamura FY, Rico-González M. Comparison of the validity and reliability of local positioning systems against other tracking technologies in team sport: A systematic review. Proc IMechE Part P: J Sports Engineering and Technology. 2021;

3. Oliva-Lozano JM, Rojas-Valverde D, Gómez-Carmona CD, Fortes V, Pino-Ortega J. Impact of contextual variables on the representative external load profile of Spanish professional soccer match-play: A full season study. European Journal of Sport Science. 2020;1–10.

4. Frencken WGP, Lemmink KAPM, Delleman NJ. Soccer-specific accuracy and validity of the local position measurement (LPM) system. Journal of Science and Medicine in Sport. 2010;13:641–5.

5. Ogris G, Leser R, Horsak B, Kornfeind P, Heller M, Baca A. Accuracy of the LPM tracking system considering dynamic position changes. Journal of Sports Sciences. 2012;30:1503–11.

6. Stevens TGA, de Ruiter CJ, van Niel C, van de Rhee R, Beek PJ, Savelsbergh GJP. Measuring Acceleration and Deceleration in Soccer-Specific Movements Using a Local Position Measurement (LPM) System. International Journal of Sports Physiology and Performance. 2014;9:446–56.

7. Linke D, Link D, Lames M. Validation of electronic performance and tracking systems EPTS under field conditions. Ardigò LP, editor. PLOS ONE. 2018;13:e0199519.

8. Luteberget LS, Spencer M, Gilgien M. Validity of the Catapult ClearSky T6 Local Positioning System for Team Sports Specific Drills, in Indoor Conditions. Front Physiol. 2018;9:115.

9. Rico-González M, Arcos AL, Rojas-Valverde D, Clemente FM, Pino-Ortega J. A Survey to Assess the Quality of the Data Obtained by Radio-Frequency Technologies and Microelectromechanical Systems to Measure External Workload and Collective Behavior Variables in Team Sports. Sensors. 2020;16.

Reviewer #2: Thank you for allowing me to review the manuscript entitled “A new approach to quantify changes-of-direction locomotion during soccer matches” which aimed to propose a new approach with direction of speed and jerk for quantifying changes-of-direction (CODs) in locomotion during soccer matches. Although interesting, several issues must be attended.

Abstract

L25. Please, include information about participants.

In general, I expect a more structured abstract. Please, follow this order and include the presented information in the section that corresponds to it:

- Background and aim

- Methods

- Results

- Conclusions

Introduction

The first paragraph must be improved. Although it shows the necessity of perform this study, the previous justification is poor, in quality and quantity.

In the second paragraph, more information about local positioning systems and its relationship with COD before to show examples. In addition, this paragraph must be improved in terms of style.

L48. Please, include the complete terms for GPS and LPS.

L57. Reference.

L 61. Reference.

L62. Reference.

Please, divide the third paragraph and include a fourth one, highlighting the necessity of perform this investigation, the main aim, and the hypothesis.

Material and methods

Participants and experimental design

This section must be divided, on one hand experimental design, an in the other hand, participants. In addition, information must be structured (following a logical order) and complete. In this sense, more information about participants is required.

Players’ coordinate data

I consider that this section is ok, but an effort by authors to facilitate its compression is required. In addition, the use of the first person style must be avoided.

Statistical analysis

More information for this section is required.

Results

Complete information is presented in this section, although improvements in style must be implemented.

Discussion

Please, for the first paragraph follow this structure: Aim – novelty of the study – main findings.

This section must be improved completely. More information about justifications (with references) and comparisons with previous studies is required.

Please, include a limitation section.

Conclusion

Please, only show the most relevant conclusions derivated of your findings, avoiding first person style.

Please, include practical applications derivated of yours finding that can be used by strength and conditioning specialists.

6. PLOS authors have the option to publish the peer review history of their article (what does this mean?). If published, this will include your full peer review and any attached files.

Reviewer #1: No

Reviewer #2: No

---

## [Decision Letter · Decision Letter 1]

26 Apr 2021

A new approach to quantify angles and time of changes-of-direction during soccer matches

PONE-D-21-04502R1

Dear Dr. Takai,

We’re pleased to inform you that your manuscript has been judged scientifically suitable for publication and will be formally accepted for publication once it meets all outstanding technical requirements.

Kind regards,

Filipe Manuel Clemente, PhD

Academic Editor

PLOS ONE

Additional Editor Comments (optional):

Reviewers' comments:

Reviewer's Responses to Questions

**Comments to the Author**

1. If the authors have adequately addressed your comments raised in a previous round of review and you feel that this manuscript is now acceptable for publication, you may indicate that here to bypass the “Comments to the Author” section, enter your conflict of interest statement in the “Confidential to Editor” section, and submit your "Accept" recommendation.

Reviewer #1: All comments have been addressed

Reviewer #2: All comments have been addressed

2. Is the manuscript technically sound, and do the data support the conclusions?

Reviewer #1: Yes

Reviewer #2: Yes

3. Has the statistical analysis been performed appropriately and rigorously? 

Reviewer #1: N/A

Reviewer #2: Yes

4. Have the authors made all data underlying the findings in their manuscript fully available?

Reviewer #1: Yes

Reviewer #2: Yes

5. Is the manuscript presented in an intelligible fashion and written in standard English?

Reviewer #1: Yes

Reviewer #2: Yes

6. Review Comments to the Author

Reviewer #1: Dear authors,

Thank you very much for addressing all my comments.

In my opinion, the article can be accepted in the current form.

Reviewer #2: Dear authors,

Thank you very much for the effort made to attend to the comments. I consider that the quality of the manuscript has been substantially improved.

7. PLOS authors have the option to publish the peer review history of their article (what does this mean?). If published, this will include your full peer review and any attached files.

Reviewer #1: No

Reviewer #2: No

---

## [Editor Report · Acceptance letter]

7 May 2021

PONE-D-21-04502R1 

A new approach to quantify angles and time of changes-of-direction
during soccer matches 

Dear Dr. Takai:

I'm pleased to inform you that your manuscript has been deemed suitable for publication in PLOS ONE. Congratulations! Your manuscript is now with our production department. 

Kind regards, 

on behalf of

Dr. Filipe Manuel Clemente 

Academic Editor

PLOS ONE